# Corticosteroids in Moderate-To-Severe Graves’ Ophthalmopathy: Oral or Intravenous Therapy?

**DOI:** 10.3390/ijerph16010155

**Published:** 2019-01-08

**Authors:** Laura Penta, Giulia Muzi, Marta Cofini, Alberto Leonardi, Lucia Lanciotti, Susanna Esposito

**Affiliations:** Paediatric Clinic, Department of Surgical and Biomedical Sciences, Università degli Studi di Perugia, 06129 Perugia, Italy; laura.penta@ospedale.perugia.it (L.P.); giulia25muzi@gmail.com (G.M.); marta.cofini@gmail.com (M.C.); alberto.leonardi88@gmail.com (A.L.); lucia.lanciotti@gmail.com (L.L.)

**Keywords:** corticosteroids, Graves’ orbitopathy, hyperthyroidism, ophthalmopathy, prednisone

## Abstract

Background: Ophthalmopathy is a rare extra-thyroid manifestation of Graves’ disease, in paediatrics. Intravenous corticosteroids are the main treatment of moderate-to-severe Graves’ orbitopathy. In this paper, we describe a moderate-to-severe active Graves’ ophthalmopathy in a child and the response to oral therapy with prednisone. Case presentation: A nine-year-old male child suffering for a few months, from palpitations, tremors, and paresthesia was hospitalized in our Pediatric Clinic. At admission, the thyroid function laboratory tests showed hyperthyroidism with elevated free thyroxine (FT4) and free triiodothyronine (FT3) levels and suppressed thyroid-stimulating hormone (TSH) levels. These findings, combined with the clinical conditions—an ophthalmologic evaluation (that showed the presence of exophthalmos without lagophthalmos and visual acuity deficiency), thyroid ultrasound, and TSH receptor antibody positivity—led to a diagnosis of Graves’ disease. Therefore, methimazole was administered at a dose of 0.4 mg/kg/day. After 4 months, thyroid function was clearly improved, with normal FT3 and FT4 values and increasing TSH values, without adverse effects. Nevertheless, an eye examination showed ophthalmopathy with signs of activity, an increase in the exophthalmos of the right eye with palpebral retraction, soft tissue involvement (succulent and oedematous eyelids, caruncle and conjunctival hyperaemia and oedema) and keratopathy, resulting from exposure. We began steroid therapy with oral administration of prednisone (1 mg/kg/day) for four weeks, followed by gradual tapering. After one week of therapy with prednisone, an eye assessment showed reduced retraction of the upper eyelid of the right eye, improvement of right eye exophthalmometry and reduction of conjunctival hyperaemia. After four weeks of therapy with prednisone, an eye assessment showed reduction of the right palpebral retraction without conjunctival hyperaemia and no other signs of inflammation of the anterior segment; after twelve weeks, an eye assessment showed a notable decrease in the right palpebral retraction and the absence of keratitis, despite persisting moderate conjunctival hyperaemia. No adverse event associated with steroid use was observed during the treatment period and no problem in compliance was reported. Conclusion: Prednisone seems a better choice than intravenous corticosteroids, for treating moderate-to-severe and active Graves’ ophthalmopathy, keeping in mind the importance of quality of life in pediatric patients.

## 1. Introduction

Graves’ ophthalmopathy, also known as Graves’ orbitopathy, thyroid eye disease, or thyroid-associated ophthalmopathy, is an inflammatory disease of the orbital tissues and the eyes [1]. Graves’ ophthalmopathy is rare in paediatric patients, and it is less common in children and adolescents than in adults. The incidence of Graves’ orbitopathy in children is 0.79–6.5 cases per 100,000 children and 1.7–3.5 cases per 100,000 population per year [2]. Graves’ orbitopathy is more common in girls than in boys and more often occurs in adolescents from 11 to 18 years old (68.2%) than in children under 11 years old (31.8%). Furthermore, Graves’ orbitopathy is less severe in children than in adults. These differences can be caused by different natural histories, genetics, and immunology of the orbitopathy. Moreover, orbital fat is most commonly affected and enlarged in children than in adults, while extraocular muscles are most commonly affected in adults [3]. This explains the absence of severe strabismus in children and the more common optic nerve compression in adults. 

Diagnosis of thyroid-associated ophthalmopathy in children and adolescents is based on the presence of ocular symptoms and signs, thyroid autoimmunity, and the exclusion of an alternative diagnosis [1]. Symptoms of Graves’ orbitopathy in children and adolescents are similar to those in adults. The most common symptoms are ocular pain, foreign body sensation, hypersensitivity to light and diplopia; while the most common signs are excessive tearing, Graefe sign or lid lag, upper eyelid retraction, proptosis, and soft tissue involvement [1,4,5]. In children and adolescents, myopathy and dysthyroid optic neuropathy may also occur in rare cases. Restricted strabismus and exposure keratopathy are serious complications that can also occur, with a higher incidence in puberty. Diagnosis of Graves’ ophthalmopathy, based only on laboratory findings of thyroid autoimmunity, has not yet been achieved and will require further studies.

TSH receptor antibodies seem responsible for the pathological changes of the eyes. The European Group on Graves’ Orbitopathy (EUGOGO) classified Graves’ orbitopathy based on severity, as mild, moderate-to-severe, and sight-threatening [6]. Anti-thyroid drugs administered for the treatment of hyperthyroidism in Graves’ ophthalmopathy cause ocular improvement in the majority of the cases [1]. Graves’ ophthalmopathy symptoms in children and adolescents often regress after restoring normal thyroid function, and the “wait and see” policy is, therefore, adequate in most of the cases. However, if there is no ocular improvement after restoring euthyroidism, pharmacological treatments with corticosteroids, usually administered intravenously, may be necessary [2,7,8,9,10,11]. In this paper, we describe a moderate-to-severe active Graves’ ophthalmopathy in a nine-year-old child and the response to oral therapy with prednisone.

## 2. Case Presentation

A nine-year-old male had complained of palpitations, tremors, and paresthesia for approximately two months. No weight loss, polyphagy, or change in mood were reported. The clinical examination showed generalized leanness associated with dry skin, rhythmic, and concited cardiac activity, fine tremors of the hands, palpable and globose thyroid, bilateral exophthalmos, retained ocular motility, and mild bilateral conjunctival hyperaemia. A direct and consensual pupillary reflex was present. Vital signs showed no fever, a heart rate of 140 beats per minute, blood pressure of 100/65 mmHg and a body temperature of 36 °C. His weight was 24.5 kg (7th percentile, −1.45 SDS), his height was 141 cm (84th percentile, 1.01 SDS), and his body mass index (BMI) was 12.32 (0.0 percentile, −3.60 SDS), according to the Italian Society for Paediatric Endocrinology and Diabetes charts [5].

At admission, electrocardiogram (ECG) showed sinus tachycardia (140 beats per minute), in the absence of other significant alterations. Thyroid function laboratory tests showed hyperthyroidism: Elevated free thyroxine (FT4) and free triiodothyronine (FT3) levels, at 2.75 ng/dL (normal value [n.v.]: 0.7–1.48), and 4.10 pg/mL (n.v.: 1.71–3.71), respectively; and suppressed and suppressed thyroid-stimulating hormone (TSH) levels, at 0.0001 µUI/mL (n.v.: 0.350–4.940). A thyroid ultrasound showed a significantly increased glandular size for age. The total volume (approximately 17.1 mL, with a right lobe of approximately 8.8 mL and a left lobe of approximately 8.3 mL) turned out to be quadrupled with respect to the age reference value: 3.1 mL +/− 1.05 mL; ultrasound also showed hypoechoic and diffusely inhomogeneous echo-structure and intensely increased vascularization of the gland. 

Ophthalmologic evaluation confirmed the presence of exophthalmos and the absence of lagophthalmos or visual acuity deficiency, with normal ocular motility, slight congestion of the eyelids (non-inflammatory oedema), and modest bulbar conjunctival hyperaemia. No diplopia was observed. The exophthalmometry detected a value of 17.00 mm for the right eye and a value of 17.5 mm for the left eye. Clinical Activity Score (CAS) was 6 and NOSPECS was IVa. These findings, combined with clinical conditions and TSH receptor antibody positivity, led to a diagnosis of Graves’ disease. Anti-peroxidase and anti-tireoglobulin antibodies were also found to be positive. Therefore, methimazole was administered at the starting dose of 0.4 mg/kg/day in two administrations, confirming that blood count and liver function were in the normal range. During hospitalization, the clinical conditions of the young patient improved, and the patient was discharged. 

After one month, when the patient returned to our clinic, weight increased from 24.4 kg to 26.5 kg, while the BMI remained under the 3rd percentile, with a value of 13.10. Blood count, liver function, albumin, and total protein, checked to assess possible adverse effects of the methimazole therapy, were normal. Thyroid function improved: FT3 was 7.32 pg/mL, FT4 was 1.65 ng/dL, and TSH was 0.005 µUI/mL. Clinical examination showed good clinical conditions and bilateral exophthalmos, greater on the right eye. CAS was 7 and NOSPECS was again IVa. The heart rate was 88 beats per minute. During the follow-up visits, at two and three months after discharge, blood count and liver function were also normal, thyroid function improved, and thyroid volume decreased; thus, the therapy was reduced to 0.27 mg/kg/day. 

After four months, thyroid function clearly improved: FT3 of 3.53 pg/mL, FT4 of 1.17 ng/dL, and TSH of 0.007 µUI/mL. Blood count and liver function were in the normal range, but the eye examination showed worsening of the ophthalmopathy, with the following signs of activity: Increase in the exophthalmos of the right eye with palpebral retraction, soft tissue involvement (succulent eyelids with associated oedema, caruncular and conjunctival hyperaemia, and oedema) and keratopathy resulting from exposure. CAS was 9 and NOSPECS Va. On the basis of the ocular findings, steroid therapy was initiated by oral administration of prednisone (1 mg/kg/day) for four weeks, and then gradually tapered. Thyroid function improved again; five months after discharge, thyroid hormones remained in the normal range: FT3 was 2.450 pg/mL, FT4 was 0.97 ng/dL, and TSH increased to 0.127 µUI/mL. A clinical examination showed a reduced thyroid volume, which was confirmed by the thyroid ultrasound that showed dimensions of 13 mL (17.1 mL at previous control) and reduced echogenicity, with non-homogeneous eco-structure, and increased vascularization of the left lobe. Therefore, therapy with methimazole was further reduced to 0.18 mg/kg/day. Thyroid function was normal, with an FT3 value of 3.59 pg/mL (n.v.: 2–4.4), and an FT4 value of 1.49 ng/dL (n.v.: 0.90–1.70), although TSH was slightly reduced with a value of 0.182 µUI/mL (n.v.: 0.270–4.200). Blood count and liver function remained normal, so therapy with methimazole was confirmed. At the last clinical follow-up, after eight months, thyroid function remained good, the exophthalmos was reduced and the patient did not have tremors. Weight increased from the first visit at admission, from 24.4 kg (3–10th percentile) to 30 kg (23rd percentile, −0.73 SDS), with a BMI of 14.37 (3rd percentile, −1.84 SDS) [5]. Furthermore, puberty started in our patient. Regarding the Graves’ ophthalmopathy trend, it has been in progressive improvement after beginning prednisone therapy. After one week of therapy, the eye assessment showed reduced retraction of the upper eyelid of the right eye, improvement of the right eye exophthalmometry and reduction of conjunctival hyperaemia. CAS was 4 and NOSPECS IIIb. After four weeks of therapy, the eye assessment showed reduction of the right palpebral retraction, without conjunctival hyperaemia, and no other signs of inflammation of the anterior segment were observed. CAS was 3 and NOSPECS IIb. After twelve weeks, the eye assessment showed a notable decrease of the right palpebral retraction and absence of keratitis and conjunctival hyperaemia. CAS was 2 and NOSPECS IVa.

Table 1 shows the clinical and laboratory parameters in our patient, at admission, and during the follow-up. Management of the case was approved by the Ethics Committee of Santa Maria della Misericordia Hospital, Perugia, Italy (2018-PED-06). The patient’s parents provided their written informed consent and the child his written assent for the management and the publication of the case report.

## 3. Discussion

This study showed a moderate-to-severe active Graves’ ophthalmopathy in a child who responded to oral therapy with prednisone. Graves’ ophthalmopathy is quite rare in pediatric age and the demonstration of efficacy and safety of oral corticosteroids in this uncommon complication, appear particularly interesting. Moreover, different therapeutic approaches have been considered. Oral corticosteroids, such as prednisolone or dehydrocortisone, have been reported in few cases in the literature, such as initial dose of 1 mg/kg/24 h for the first week and then tapering to 10 mg/24 h for three months, or an initial dose of 40–60 mg/24 h and then tapering to 15 mg daily for four or six months [12]. Usually, intravenous corticosteroids are recommended for moderate-to-severe active Graves’ ophthalmopathy. The effectiveness of pulse therapy with intravenous methylprednisolone administered once a week for six weeks and then with pulses administered weekly for six weeks has been shown [13]. Another regimen consisted of administration of 15 mg/kg intravenous methylprednisolone for four cycles, each of which consisted of two infusions on alternate days at two-week intervals, followed by 7.5 mg/kg for a further four cycles [14]. Another option is the administration of methylprednisolone pulses in 24 h over three to five successive days, depending on the severity of the orbitopathy and the response to treatment, which can be appreciated after 7–14 days of therapy [15]. These regimens are mostly effective and usually safe [16,17], although cases of fatal liver failure have been described after high-dose intravenous steroids [12]. It is also known that prolonged oral prednisone administration can cause adverse effects, such as weight gain, immune suppression, and growth failure [18,19]. Nevertheless, we decided, in agreement with the family (the parents of this child did not give their consent to intravenous therapy as the first-line approach), to start oral prednisone, with the aim of adopting the therapeutic choice best suited for the patient’s lifestyle. We wanted to ensure the best possible quality of life for our patient, reserving the possibility of using intravenous therapy in case of worsening ocular conditions. No adverse event associated with steroid use was observed during the treatment period and no problem in compliance was reported. 

## 4. Conclusions

This case showed that it is possible to successfully employ oral corticosteroids for the treatment of moderate-to-severe active Graves’ ophthalmopathy, in pediatric age, keeping in mind the importance of quality of life for patients in the first years of life, according to their lifestyle and family needs.

## Figures and Tables

**Table 1 ijerph-16-00155-t001:** Clinical and laboratory parameters in a nine-year-old child with moderate-to-severe active Graves’ ophthalmopathy treated with methimazole and oral prednisone.

Parameter	Timeline								
	Admission	1 m	2 m	3 m	4 m	5 m	6 m	8 m	12 m
Graves’ ophthalmopathy	Mild EP and conjunctival hyperemia				Worsening EP, palpebral retraction, soft tissue involvement, keratopathy and conjunctival hyperemia	Reduced EP		Reduced EP, moderate conjunctival hyperemia	No EP
TSH (µUI/mL) (n.v. 0.27–4.2)	0.0001	0.005	0.005	0.005	0.007	0.127	0.182	1.11	1.730
FT3 (pg/mL) (n.v. 2–4.4)	4.10	7.32	6.51	5.62	3.53	2.450	3.59	4.26	4.28
FT4 (ng/dL) (n.v. 0.9–1.7)	2.75	1.65	1.64	1.58	1.17	0.97	1.49	1.3	1.30
Hearth rate (bpm)	140	88						80	74
Blood pressure (mmHg)	100/65	108/67				98/65		103/58	101/75
Therapy with methimazole (mg/kg/die)	0.4	0.4	0.27	0.27	0.27	0.18	0.08	0.08	0.17
Therapy with prednisone					1 mg/kg/die	0.45 mg/kg/die	Stopped		

FT4: free thyroxine; FT3: free triiodothyronine; TSH: suppressed thyroid-stimulating hormone; m: months; n.v.: normal values; EP: exophthalmos.

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
