# Peer review of "Corticosteroids in Moderate-To-Severe Graves’ Ophthalmopathy: Oral or Intravenous Therapy?"

_ijerph, 2019, doi:10.3390/ijerph16010155_

Round 1
Reviewer 1 Report
Penta et al describe a case of GO in an 11 years old child treated with peroral corticosteroids entitled: Corticosteroids in moderate-to-severe Graves’ ophthalmopathy: oral or intravenous therapy?
GO are uncommon in children with Graves’ disease, and it can be discussed if a case in an 9-year-old boy deserve a case- presentation.
The traditional treatment modality for GO was for many years peroral corticosteroid. The EUGOGO guidelines (ref 6) demonstrated that intravenous treatment was more effective and had less side-effect and therefore the recommendation has been iv treatment. But still some time peroral therapy is used.
Therefore, it is not surprising that there is effect of therapy in this case.
In the case I do miss the following information:
The reason for giving the treatment peroral instead of intravenious – they use QoL as argumentation – one could argue for using Iv on the same argumentation related to possible side effect.
Description of side effect from peroral corticosteroids in the actual case.
A describing of clinical activity score (CAS) and Nospect during the treatment
Author Response
Comments and Suggestions for Authors
Penta et al describe a case of GO in an 11 years old child treated with peroral corticosteroids entitled: Corticosteroids in moderate-to-severe Graves’ ophthalmopathy: oral or intravenous therapy?
GO are uncommon in children with Graves’ disease, and it can be discussed if a case in an 9-year-old boy deserve a case- presentation.
Re: We have tried to explain better than before in the Discussion the importance of the publication of this case due to the request of the family that refused iv therapy as well as the efficacy and safety of the treatment (pp. 1 and 5).
The traditional treatment modality for GO was for many years peroral corticosteroid. The EUGOGO guidelines (ref 6) demonstrated that intravenous treatment was more effective and had less side-effect and therefore the recommendation has been iv treatment. But still some time peroral therapy is used.
Therefore, it is not surprising that there is effect of therapy in this case.
Re: We have included data on safety (pp. 1 and 5).
In the case I do miss the following information:
The reason for giving the treatment peroral instead of intravenious – they use QoL as argumentation – one could argue for using Iv on the same argumentation related to possible side effect.
Description of side effect from peroral corticosteroids in the actual case.
Re: The family refused iv therapy and asked to begin oral corticosteroids. The efficacy and safety of the treatment have been reported (pp. 1 and 5).
A describing of clinical activity score (CAS) and Nospect during the treatment.
Re: Added (pp. 2 and 3).
Reviewer 2 Report
The present manuscript is a clinical case of a 9-year old boy with hyperthyroidism (Grave's disease). The patient presented with exophthalmos. The patient was successfully treated with methimazole, and after four months, thyroid function was improved, with normal FT3 and FT4 20 values and increasing TSH values without adverse effects. However, no improvement in eye pathology was found. Therefore, the patient was treated with oral prednisone for four weeks. After the first week of treatment, the improvement was evident. After 4 weeks of therapy with prednisone, an eye assessment showed a reduction of the right palpebral retraction without conjunctival hyperemia and no other signs of inflammation of the anterior segment, and after 12 weeks, they found a significant decrease in the right palpebral retraction and the absence of keratitis, although, the patient was still suffering from conjunctival hyperemia. The authors conclude that oral prednisone is a good therapeutic choice to treat Graves’ ophthalmopathy.
This is a well-described case report.
The manuscript will benefit by adding pictures of the pathological findings. Including pictures will provide stronger evidence of the beneficial effects of oral prednisone vs. i .v. Administration of corticosteroids.
I do not see any major issues other than the need of including pictures of the pathology they described. Also, perhaps adding a table with more detailed clinical data will be important. I am sure authors have these data.
Minor comments:
The manuscript will benefit by including pictures of the pathological findings. Including pictures will provide stronger evidence of the beneficial effects of oral prednisone vs. i .v. Administration of corticosteroids.
Author Response
The present manuscript is a clinical case of a 9-year old boy with hyperthyroidism (Grave's disease). The patient presented with exophthalmos. The patient was successfully treated with methimazole, and after four months, thyroid function was improved, with normal FT3 and FT4 20 values and increasing TSH values without adverse effects. However, no improvement in eye pathology was found. Therefore, the patient was treated with oral prednisone for four weeks. After the first week of treatment, the improvement was evident. After 4 weeks of therapy with prednisone, an eye assessment showed a reduction of the right palpebral retraction without conjunctival hyperemia and no other signs of inflammation of the anterior segment, and after 12 weeks, they found a significant decrease in the right palpebral retraction and the absence of keratitis, although, the patient was still suffering from conjunctival hyperemia. The authors conclude that oral prednisone is a good therapeutic choice to treat Graves’ ophthalmopathy.
This is a well-described case report.
Re: Thank you very much for the appreciation of our manuscript.
The manuscript will benefit by adding pictures of the pathological findings. Including pictures will provide stronger evidence of the beneficial effects of oral prednisone vs. i .v. Administration of corticosteroids.
I do not see any major issues other than the need of including pictures of the pathology they described. Also, perhaps adding a table with more detailed clinical data will be important. I am sure authors have these data.
Re: Unfortunately, the parents did not provide us the consent to publish the pictures. However, details on CAS and NOSPECT have been included (pp. 2 and 3).
Minor comments:
The manuscript will benefit by including pictures of the pathological findings. Including pictures will provide stronger evidence of the beneficial effects of oral prednisone vs. i .v. Administration of corticosteroids.
Re: Due to the absence of parents’ consent, details on CAS and NOSPECT have been included (pp. 2 and 3).
Reviewer 3 Report
IJERPH-402104
Suggestions/Comments
The authors should well explain that How and Why was the regimen they performed being the best therapeutic protocol for the case. What is (are) the NEW/NOVEL concept (s) and/or finding (s) from the case? What is (are) the specific point (s) of the case? The authors should interpret and discuss further in the Discussion section.
Author Response
The authors should well explain that How and Why was the regimen they performed being the best therapeutic protocol for the case. What is (are) the NEW/NOVEL concept (s) and/or finding (s) from the case? What is (are) the specific point (s) of the case? The authors should interpret and discuss further in the Discussion section.
Re: As suggested, further details on the novelty of the case have been included.
Round 2
Reviewer 1 Report
I still do not find any new aspects in this case - evaluated on the CAS and NOSPECT score it was a severe case of GO in a child. If the parent were properly informed of the possible effect versus side effect on iv contra po corticosteroid - I do think that most parents would accept the the iv treatment.
These parents chose the second best treatment and still the boy recovered - is that the case?